

# Plume Propagation Direction Determination with $SO_2$ Cameras

Angelika Klein[1], Peter Lübcke[1], Nicole Bobrowski[1], Jonas Kuhn[1], and
Ulrich Platt[1]

[1]Institute of Environmental Physics, University of Heidelberg

*Correspondence to:* Angelika Klein (angelika.klein@iup.uni-heidelberg.de)

**Abstract.** $SO_2$ cameras are becoming an established tool for measuring sulphur dioxide ($SO_2$) fluxes in volcanic plumes with good precision and high temporal resolution. The primary result of $SO_2$ camera measurements are time series of two-dimensional $SO_2$ column density distributions (i.e. $SO_2$ column density images). However, it is frequently overlooked that in order to determine the

correct $SO_2$ fluxes, not only the $SO_2$ column density, but also the distance between the camera and the volcanic plume has to be precisely known. This is because cameras only measure angular extensions of objects while flux measurements require knowledge of the spatial plume extension. The distance to the plume may vary within the image array (i.e. the field of view of the $SO_2$ camera) since the plume propagation direction (i.e. the wind direction) might not be parallel to the image

plane of the $SO_2$ camera. If the wind direction and thus the camera-plume distance is not well known, this error propagates into the determined $SO_2$ fluxes and can cause errors exceeding 50%. This is a source of error which is independent of the frequently quoted (approximate) compensation of apparently higher $SO_2$ column densities and apparently lower plume propagation velocities at non-perpendicular plume observation angles.

Here, we propose a new method to estimate the propagation direction of the volcanic plume directly from $SO_2$ camera image time series by analysing apparent flux gradients along the image plane. From the plume propagation direction and the known location of the $SO_2$ source (i.e. volcanic vent) and camera position the camera-plume distance can be determined. Besides being able to de-

termine the plume propagation direction, and thus the wind direction in the plume region, directly from $SO_2$ camera images, we additionally found, that it is possible to detect changes of the propagation direction at a time resolution on the order of minutes. In addition to theoretical studies we applied our method to $SO_2$ flux measurements at Mt. Etna and demonstrate that we obtain considerably more precise (up to a factor of 2 error reduction) $SO_2$ fluxes. We conclude that studies on $SO_2$

flux variability become more reliable by excluding the possible influences of propagation direction variations.



## 1 Introduction

Prediction and monitoring of volcanic events is highly desirable. Besides conventional methods, like seismicity or deformation measurements, continuous monitoring of volcanic gas emissions is a still

relatively new method for predicting volcanic eruptions. The four most common changes in volcanic behaviour preceding an eruption are earthquakes, deformation, thermal anomalies, and an increase in degassing of the volcano. Moreover, not only an increase in its degassing behaviour but also a change in composition of the volcano's degassing can be an indicator of an imminent eruption (see e.g. Bobrowski et al., 2015).

For short term as well as long-term monitoring of volcanic degassing behaviour in-situ as well as remote-sensing techniques have been developed. While in-situ techniques, such as alkaline traps and MultiGAS (Noguchi and Kamiya, 1963,Aiuppa et al., 2007) have been successfully applied, remote-sensing techniques have the particular advantage that they can be applied from a safe distance. Remote-sensing started with the correlation spectrometer (COSPEC, Moffat and Millan, 1971

and Stoiber et al., 1983) but more recently the differential optical absorption spectroscopy (DOAS) technique (Platt and Stutz, 2008) is applied at volcanoes. Long-term remote-sensing monitoring of the $SO_2$ emission rate (e.g. by the Network for Observation of Volcanic and Atmospheric Change (NOVAC), Galle et al., 2010) provides insights in the standard behaviour of each individual volcano and deviations from the normal activity can be used to predict eruptions. More recently, the SO2

camera (e.g. Mori and Burton, 2006) that can record two dimensional $SO_2$ column density distributions allowed unprecedented insight into chemical and dynamic processes in volcanic plumes. Future developments promise further improvements (Platt et al., 2015).

The $SO_2$ camera is a UV sensitive camera utilizing one or more band-pass interference filters to

measure the optical density (OD) of $SO_2$. One of those interference filters has a central transmission wavelength at about 310 - 315 nm. This filter is used to determine the light extinction mainly due to $SO_2$ and aerosols. The light extinction due to aerosol exhibits a broad band structure when compared to the narrow band structure caused by the light attenuation due to $SO_2$. Therefore, a second filter is applied with a center wavelength of approximately 330 nm, where the $SO_2$ absorption is negligible,

but which is close enough to cause only small changes in light extinction by aerosol (Lübcke et al., 2013). From the logarithm of the (suitably normalized) pixel-per-pixel ratio of two images taken through either filter, images of the $SO_2$ OD can be calculated. The $SO_2$ OD in turn is proportional to the $SO_2$ column density along the line of sight.

The propagation velocity of the plume and the distance between the plume and the camera are two

important variables to determine the $SO_2$ emission rate from volcanoes using imaging data. Usually the apparent propagation velocity (i.e. the angular velocity) of the plume can be derived directly from the camera image series. For that purpose one correlates two integrated transects of the trace gas slant column density images of the moving plume, and determines the time lag between the two



signals (McGonigle et al., 2005). One can determine the velocity of the plume from the time lag, the
angular distance between the two image columns, and the distance of the plume. While this method
is simple to implement it only provides a spatial and temporal mean propagation velocity that ne-
glects for example turbulence or propagation velocity variations over the extension of the plume.
A more detailed plume velocity determination can be achieved using optical flow algorithms (Kern
et al., 2015b). These algorithms determine the displacement of image intensity values for each pixel
from one frame to the next frame, thus giving a detailed spatial and temporal plume velocity estima-
tion if the direction of the plume is known.

In any case an important prerequisite for the determination of an absolute trace gas flux values
is the precise knowledge of the distance between plume and observing instrument (usually the $SO_2$
camera). This distance is usually more difficult to (precisely) determine than it is generally assumed:
While the geographic locations of the volcanic gas source (i.e. usually the crater) and the position of
the instrument are almost always precisely known, the plume propagation direction (like the plume
velocity) is not. It is advantageous to know the propagation direction of the plume to achieve a
good estimation of the plume distance. This usually requires additional measurements, which are
often hard to make at volcanoes due to the limited infrastructure. This paper is about the possibility
to determine the plume propagation direction itself from a time series of $SO_2$ camera images of a
volcanic plume.

## 2   Theory

The trace gas flux $\Phi$ is approximated from 2D imaging data following the equation

$$\Phi = \boldsymbol{v} \cdot \sum_i h_i \cdot S_i \tag{1}$$

Here, $\boldsymbol{v}$ is the propagation velocity of the plume perpendicular to the viewing direction, $h_i$ is a side
length of a pixel at the distance of the plume and $S_i$ denotes the $SO_2$ column densities of each re-
spective pixel.

If the volcanic $SO_2$ plume moves within the image plane, the camera captures a scaled image of the
Field of View (FOV) of the camera image, with a scaling factor dependent on the plume distance.
Thus, the height of the plume and its propagation velocity can be easily calculated once the plume
distance is known. In a simplified approach neglecting radiation transport issues (as described e.g.
by Kern et al., 2010) and assuming a homogeneous $SO_2$ distribution within the plume, the column
densities $S_i$ depend linearly on the length of the light path through the plume. For a cylindrically
symmetric plume moving parallel to the object plane, the detector pixels at the center of the plume
capture column densities corresponding to the $SO_2$ concentration integrated along the plume diam-
eter, while the detector pixels towards the border of the plume capture the light-path along secants



of the plume. Since the secants are not exactly parallel to the radius this causes an overestimation
of the measured SO$_2$ column densities towards the edges of the detector. Furthermore, if the plume
is inclined (by the angle $\alpha$, see Fig. 1) with respect to the image plane, deviations in the SO$_2$ flux
determination of the plume will occur even in the center of the image plane. In the following sections
different approaches to take the geometry into account during the calculation of SO$_2$ emission rates
will be discussed. The angle between the image plane and the tilted plume will be referred to as
inclination angle $\alpha$. The inclination of the plume changes all the measured variables in Eq. 1.

**2.1   Small FOV Angle Approach**

In a first simplified approach for a small FOV angle of a few degree, the inclination deviations are
negligible (below 10 % change in SO$_2$ flux at $\alpha$ smaller than 2 degrees). Figure 1 shows a schematic
sketch of the geometry of the setup of an inclined plume. The actual plume extension in x-direction
$x_R$ of a tilted plume imaged with the SO$_2$ camera is longer than the apparent plume extension $x_M$
projected on the image plane. It can be calculated as

$$x_R = \frac{x_M}{\cos\alpha} \qquad (2)$$

The true plume velocity $v_R$ (in x-direction) depends linearly on the plume extension ($v_R = \frac{x_R}{t} = \frac{x_M}{\cos(\alpha)t} = \frac{v_M}{\cos(\alpha)}$). In contrast to the apparent underestimation of the plume velocity, the measured
column densities $S_M$ for an inclined SO$_2$ plume are larger than the perpendicular column densities
$S_R$. The column density correction follows the equation

$$S_R = S_M \cdot \cos\alpha \qquad (3)$$

The column densities depend linearly on the light path $s$ through the plume in a first order approx-
imation for a homogeneous plume with an SO$_2$ concentration $c$ ($S = c \cdot s$). Therefore, Eq. 3 can be
rewritten as

$$s_R = s_M \cdot \cos\alpha \qquad (4)$$

In this first assumption (FOV $\leq 2°$) the deviations in the velocity and in the column density would
cancel each other out in the flux calculation (see Eq.1) as already noted by Mori and Burton (2006).
Only the apparent plume diameter $h_M$ (i.e. the vertical extension of the plume in the direction of
the y-axis) would be affected and thus deviate from the true plume diameter $h_R$, since the actual
distance of the plume differs from the assumed distance which causes a wrong scaling of the plume
diameter on the image plane.

$$h_R = h_M + \frac{1}{2} \cdot x_M \cdot \tan\alpha \qquad (5)$$

It should be noted, that the $x_R$ and $s_R$ over- and underestimations nearly cancel each other out for
SO$_2$ cameras with a small FOV angle but also for a chosen small FOV angle within the large FOV



angle of an $SO_2$ camera. However, the distance of the plume still needs to be known to determine the correct plume diameter and thus also the information about the propagation direction of the plume is a necessary prerequisite even in this approach.

### 2.1.1 Large FOV Angle Approach

Usually, $SO_2$ cameras have a relatively large FOV angle $\gamma$ (typically several 10 degrees, Fig. 2). Therefore, a more realistic approach includes the angular aperture of the FOV in the determination of the variation of the variables in Eq. 1.

For FOV angles of the $SO_2$ camera larger than 2 degree, the apparent plume extension in x-direction and also the column densities are affected in a way that is different from the approach before if the

140 plume is tilted with respect to the image plane (i.e. at non-zero alpha).

The plume length deviation equation (Eq. 2) changes if the FOV projection is taken into account. Additionally to the deviations $x_K$ of an orthographic projection (every distance is projected with the same magnification factor, see gray section in plume length $x_R$ in Fig. 2), the perspective projection leads to an addition of a length $x'_K$ (see red section in plume length $x_R$ in Fig. 2) for a plume moving

away from the observer (alpha > 0) and subtraction of $x'_K$ (i.e. $x'_K$ becoming negative) if the plume moves towards the observer (apha < 0). The additional length $x'_K$ can be calculated with the law of sines.

$$x_K = \frac{x_M}{\cos\alpha} \tag{6}$$

$$\frac{x'_K}{\sin\gamma} = \frac{q}{\sin(90° - \alpha - \gamma)} = \frac{x_M \cdot \tan\alpha}{\sin(90° - \alpha - \gamma)} \tag{7}$$

$$\rightarrow x'_K = \frac{x_M \cdot \sin\gamma\tan\alpha}{\sin(90° - \alpha - \gamma)} \tag{8}$$

$$x_R = \frac{x_M}{\cos\alpha} \cdot \left(1 + \frac{\sin\gamma\sin\alpha}{\sin(90° - \alpha - \gamma)}\right) \tag{9}$$

Equation 9 can be rewritten as Eq. 10. Besides the scaling with $\cos\alpha$ from the small FOV angle approach (see Eq.2), an additional term describes the influence of the FOV angle $\gamma$ on $x_R$.

$$x_R = \frac{x_M}{\cos\alpha}\left(1 - \frac{\tan(\gamma)\tan(\alpha)}{1 + \tan(\gamma)\tan(\alpha)}\right) \tag{10}$$

Additionally, the length of the slant beam through the plume is not only dependent on the tilt angle $\alpha$ but also on the FOV angle $\gamma$ of the respective pixel following equation

$$s_R = s_M \cdot \cos\left(\alpha + \frac{\gamma}{2}\right) \tag{11}$$





The distance of the plume also changes for every FOV angle in dependence of the inclination of the plume.

$$d_R = \frac{r}{\tan\left(\frac{\gamma}{2}\right)} = \tan\alpha \cdot r + d_M \tag{12}$$

$$r = \tan\left(\frac{\gamma}{2}\right)\tan\alpha \cdot r + \tan\left(\frac{\gamma}{2}\right) \cdot d_M \tag{13}$$

$$r \cdot \left(1 - \tan\left(\frac{\gamma}{2}\right)\tan\alpha\right) = \tan\left(\frac{\gamma}{2}\right) \cdot d_M \tag{14}$$

$$r = \frac{\tan\left(\frac{\gamma}{2}\right) \cdot d_M}{1 - \tan\left(\frac{\gamma}{2}\right)\tan\alpha} \tag{15}$$

$$d_R = \frac{\tan\left(\frac{\gamma}{2}\right) \cdot d_M}{\tan\left(\frac{\gamma}{2}\right) \cdot \left(1 - \tan\left(\frac{\gamma}{2}\right)\tan\alpha\right)} \tag{16}$$

$$d_R = \frac{d_M}{1 - \tan\left(\frac{\gamma}{2}\right)\tan\alpha} \tag{17}$$

Figure 3 shows the deviations of the tilt corrected values of the variables (with index $R$) from the measured values of the variables (indicated by the index $M$) in dependence of the inclination angle $\alpha$. The distance of the plume in the midst of the FOV is known but since the direction of the plume is not known, also the distances in other positions of the image are not known. The camera's FOV angle is chosen as $24°$ which is in the range of commonly used $SO_2$ camera FOV today (Kern et al., 2015a). The graphs show the deviations for half of the image plane from its center to an angle $\frac{\gamma}{2}$ of $12°$ where half of the image plane is chosen to be a detector (consisting only of one large pixel). An orthographic projection leads to the blue lines in Fig. 3. The red lines in Fig. 3 represent the deviations due to a perspective projection that is more common in $SO_2$ camera measurement setups. Figure 4 shows the combined deviations that would influence the flux determination.

These calculations cover half of the actual FOV angle of the camera. If the plume is inclined with the angle $\alpha$ towards the image plane, it is tilted by a negative angle $-\alpha$ for the respective other half of the FOV. Therefore the over- and underestimations of the actual $SO_2$ flux differ on both sides of the field of view.

Usually the $SO_2$ camera detectors consist of several hundred pixels. Equation 18 represents the deviations in the plume length and therefore in the plume propagation velocity if the FOV angle is divided in a finite absolute number of pixels $p$ for every pixel $i$ in $p$.

$$x_R(i, x_M) = \frac{x_M}{\cos\alpha}\left[1 + i^2\frac{\tan\gamma\tan\alpha}{p - i\tan\gamma\tan\alpha} - (i-1)^2\frac{\tan\gamma\tan\alpha}{p - (i-1)\tan\gamma\tan\alpha}\right] \tag{18}$$

Equation 19 represents the deviations in the measured column densities if the FOV angle is divided in a finite absolute number of pixels $p$ for every pixel $i$ in $p$.

$$s_R(i, s_M) = s_M \cdot \cos\left[\alpha + \tan^{-1}\left(\frac{i-1}{p}\tan\gamma\right) - \frac{1}{2} \cdot \tan^{-1}\left(\frac{i}{p}\tan\gamma\right)\right] \tag{19}$$



Equation 20 represents the deviations in the measured distance for every pixel $i$ for a tilted plume.

$$d_R(i, d_M) = \frac{d_M}{1 - \tan\alpha \tan\left[\tan^{-1}\left(\frac{i-1}{p}\tan\gamma\right) - \frac{1}{2}\cdot\tan^{-1}\left(\frac{i}{p}\tan\gamma\right)\right]} \tag{20}$$

If we want to determine the plume propagation direction, we can measure the $SO_2$ flux for a given distance in different positions of the plume. If the mean flux is the same over a given time period for the different positions in the plume, the plume lies within the image plane. Otherwise we observe an apparent gradient in the measured fluxes that however contains the plume propagation direction information of the plume. Dividing the mean measured fluxes with the respective deviations for the investigated pixel columns for every possible tilt angle $\alpha$ and minimizing the observed gradient yields the information about the mean plume propagation direction during the respective time period. The conservation of the mean $SO_2$ flux assumption can be made since the mean lifetime of $SO_2$ in the troposphere is in the order of several days (Eisinger and Burrows, 1998) while typical $SO_2$ camera time series measurements are made at plumes with an age of seconds or minutes after emission.

Figure 5 shows the deviation in each measurement variable separately for an $SO_2$ camera with again a typical FOV of $24°$, while Fig. 6 shows the combined deviation of the flux measurement due to the perspective influence on the three variables. For an $SO_2$ camera with a typical FOV of $24°$ the deviations in the ratio $\frac{x_R}{x_M}$ easily exceeds 50 % at plume direction tilts of $> 30$ degrees. As a consequence, the $SO_2$ flux deviation already exceeds ten percent in parts of the $SO_2$ camera images for a plume tilt angle larger than $15°$.

Equations 18, 19 and 20 are defined for the case that the best known distance between the observer and the plume is in the center of the FOV. If the best known distance is not in the center of the FOV, the equations can be adjusted since the distance correction $d_R$ and the inclination velocity correction $x_R$ change (see also Fig. 7).

With $n$ as the number of pixels that the known position is shifted to the side of the FOV, we can derive the new distance of the center of the FOV as

$$d'_M = d_M + d_P \tag{21}$$

$$d_P = n \cdot x_M \cdot \tan\alpha = \frac{n}{p} \cdot d_M \cdot \tan\gamma \tan\alpha \tag{22}$$

$$d'_M = dM \cdot \left(1 + \frac{n}{p}\tan\alpha\tan\gamma\right) \tag{23}$$

Accordingly the new measured lengths of the pixels $x'_M$ can be calculated as:

$$x'_M = x_M \cdot \left(1 + \frac{n}{p}\tan\alpha\tan\gamma\right) \tag{24}$$

## 3 Application

We used the considerations developed in Sect. 2 (together with the usually well justified assumption of a constant $SO_2$ flux) to design an algorithm which allows to determine the wind direction directly





from SO$_2$ camera plume images without the need for further data or assumptions. The new algorithm has been applied to an SO$_2$ camera measurement data set taken at Mount Etna, Sicily on 9th July 2014. Not only the possibility of the inclination angle estimation but also the possibility of the observation of a wind direction change were investigated. Figure 8 shows the geometry of the data
set.

### 3.1 Plume propagation direction determination

Figure 9 shows the SO$_2$ fluxes at three different positions in the FOV of the SO$_2$ camera for a measurement data set taken at Mount Etna. The upper panel shows the SO$_2$ flux for each of these positions not corrected for inclination. The lower panel shows the fluxes corrected for the inclina-
tion. Parallel to the SO$_2$ camera measurements, measurements were taken by a DOAS instrument mounted on a car and looking to the zenith by traversing underneath the plume (see e.g. McGonigle et al., 2002, Galle et al., 2003). The center of the plume can be found by evaluating the SO$_2$ CD and determining the location with the maximum values. Thus, the wind direction could be estimated, giving an inclination of the plume of about 38 degree. Figure 10 shows the fluxes at seven different
positions in the camera image from the data set taken at Etna on 9th of July 2014. The SO$_2$ emission rate was calculated assuming different plume inclination angles. Figure 10 shows that the SO$_2$ emission rates are nearly the same if the plume is tilted about 40 degree in the backward direction with an uncertainty of $\pm 5$ degree. This result is nicely comparable with the result from the traverse measurements.

### 3.2 Real-Time Tracking of Changes in the Wind Direction

If there are changes in the propagation direction of the plume during SO$_2$ camera measurements, it is possible to detect these changes on time scale of minutes. Figure 11 shows a change in the ratio between the apparent SO$_2$ flux determined in two different positions of the plume within the FOV
of the camera. During 2 hours measurement between 11:32 - 13:23, on 9th July 2014, the wind direction was stable for about one hour (A), then the inclination angle changed about 20 degree, which we attribute to a change of the wind direction (B) until the new wind direction stabilized in (C).

### 4 Conclusions

We showed that an inclined plume causes apparent spatial flux gradients in the SO$_2$ camera measurement images. The frequently implicitly (e.g. Smekens et al., 2015) or explicitly (e.g. Mori and Burton, 2006) assumed compensation effect only occurs at very small inclination angles ($< 15$ degree) or small FOV ($< 2$ degree). For an SO$_2$ camera with an FOV angle of 24 degree a tilt angle of



15 degree already causes flux deviations larger than 10 percent in parts of the $SO_2$ camera's images

for evaluations relying on the compensation effect. However, these gradients are unambiguous for every possible inclination angle of the volcanic plume with respect to the image plane of the $SO_2$ camera. Therefore, they can not only be corrected but also be used to determine the direction of the plume (i.e. the wind direction at the location of the plume). On longer time scales, even the change in the mean wind direction can be observed. On the other hand, if these errors in plume inclination

are ignored they can give rise to erroneous observations of fluxes, in particular fake flux changes with plume age or over time can occur. If these changes in flux are attributed to chemical processes in plumes (e.g. $SO_2$ oxidation) or of volcanic degassing patterns, wrong conclusions with respect to chemical processes in volcanic plumes or wrong interpretation on degassing behaviour may be drawn.

*Acknowledgements.* The authors want to thank Sebastian Illing and Marco Huwe, who built the $SO_2$ camera set up that was used to acquire the measurement data sets at Mount Etna. Further, the authors thank for the financial support from the DFG project "DFG BO 3611/1-2.



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



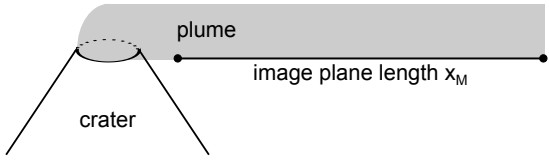

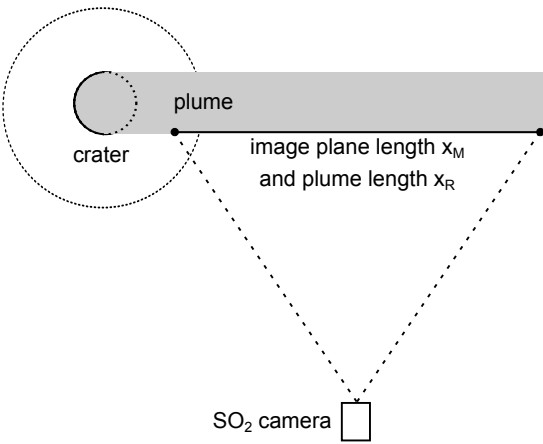

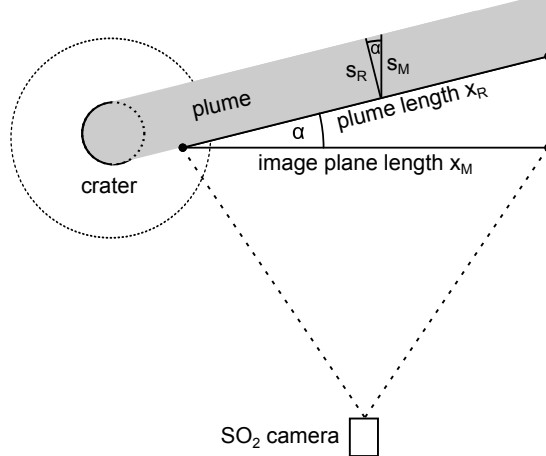

**Figure 1.** Schematic view on the influence of the inclination of the plume on the measured variables for the $SO_2$ flux determination for an $SO_2$ camera with a small FOV angle. a) side view on the volcanic plume and b) top view of a volcanic plume parallel to the image plane, c) top view of a plume inclined with respect to the image plane.




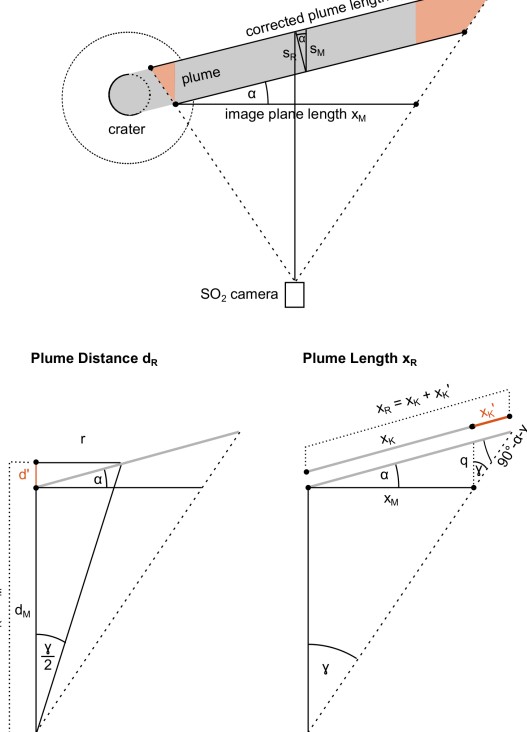

**Figure 2.** Schematic sketch of the influence of a large FOV angle on the deviation of the plume length $x_R$ from the assumed plume length $x_M$ and on the plume distance $d_R$ from the assumed plume distance $d_M$. The additional distance the plume travels in comparison to the case of a plume moving away from the camera ($\alpha = 0$) is marked in red. Here, only the case of a plume moving away from the camera ($\alpha > 0$) is shown.





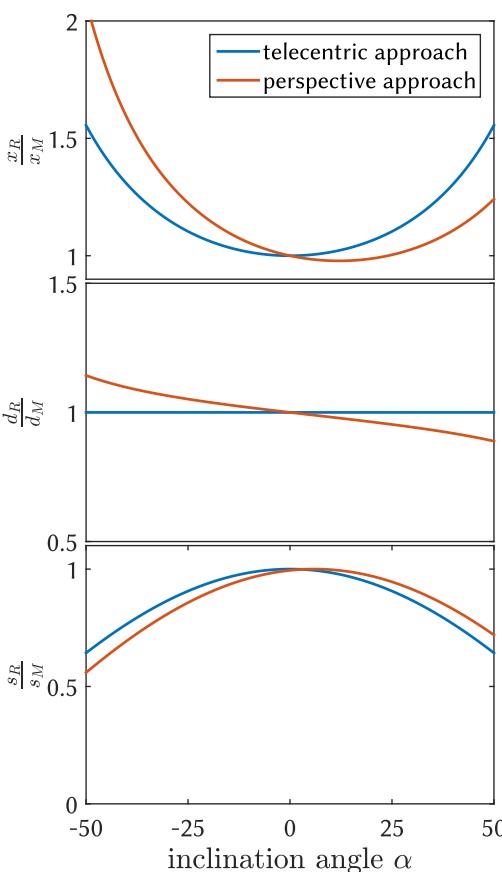

**Figure 3.** Mean deviations of the three variables plume extend in x-direction ($x_R$), plume diameter ($d_R$), and CD ($s_R$) used in the flux determination for the right half of the image plane of an $SO_2$ camera with an FOV angle of 24 degree (for the left half of the detector the inclination deviations would be vice versa). The blue lines show the ratio between the ground truth (i.e. the geometric accurate) variables and the measured variables for a telecentric (an orthographic projection where the apparent size does not depend on the distance) approach. The red lines show the same ratio for a perspective approach.




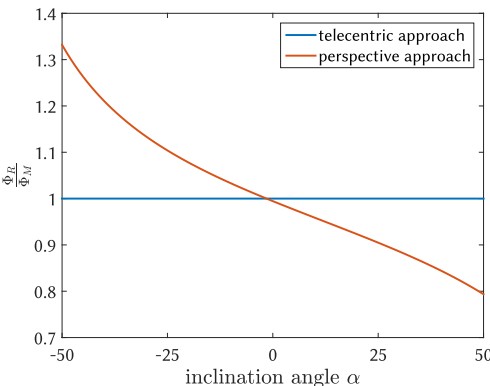

**Figure 4.** Combined deviations of the three variables of the flux determination for the right half of the image of an SO$_2$ camera with an FOV angle of 24 degree.

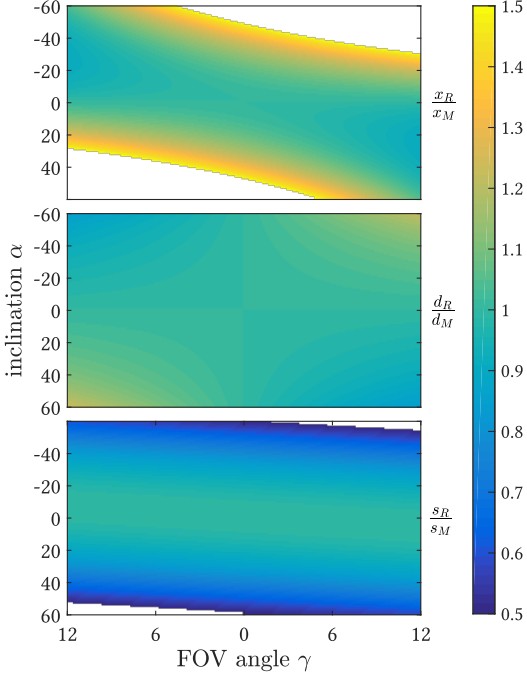

**Figure 5.** Ratio of the real variables to the measured variables of the velocity (upper panel), plume distance or diameter respectively (middle panel) and column densities (lower panel) for an SO$_2$ camera with an FOV angle of 24 degree. The relative deviations of distance are the same as for the diameter ($\frac{h_R}{h_M} = \frac{d_R}{d_M}$). Relative deviations larger than 0.5 from the measured data are shaded white.





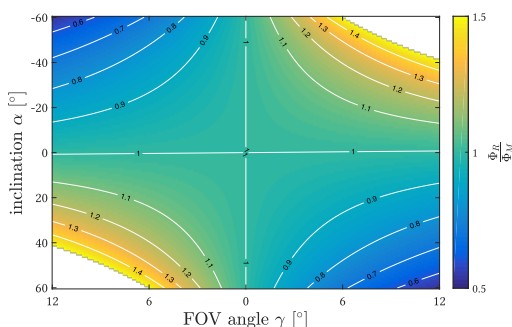

**Figure 6.** Deviation of the total true flux from the measured flux (with no inclination assumed) in dependence of the FOV angle of the $SO_2$ camera and the inclination angle $\alpha$. A perspective imaging of a plume with unknown inclination can lead to wrong flux estimations. Deviations larger than 0.5 from the measured data are shaded white. White stripes show the 0.1 steps of the deviations.

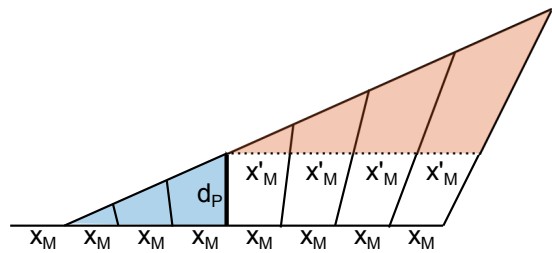

**Figure 7.** Schematic drawing of the shift of the best known distance towards the border of the image array. The equations for the perspective correction can be adapted to the respective position of the best known distance.

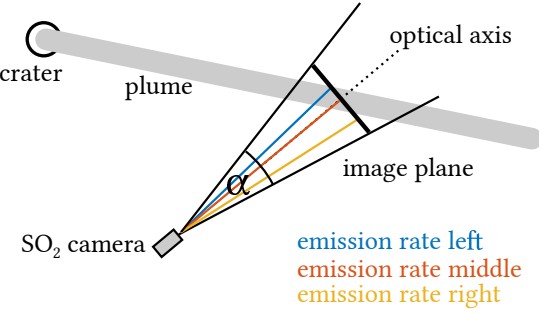

**Figure 8.** Map of the geometrical setup of the measurement data set. The inclination of the plume is 38 degree with respect to the image plane. The positions in the FOV used for the emission rate determination are colored respectively.





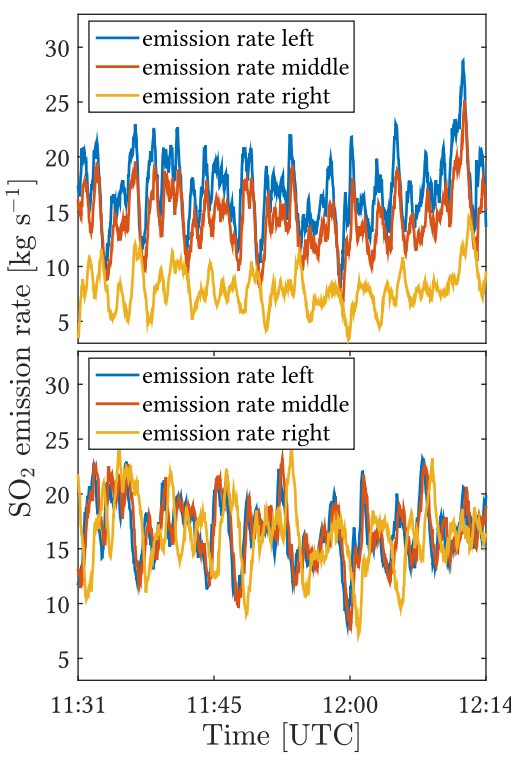

**Figure 9.** Deviations of the SO$_2$ fluxes of three different cross sections through the plume. These apparent deviations are caused by the unknown inclination of the plume with respect to the image plane. This measurement set was taken at Mount Etna on the 9th of July 2014.




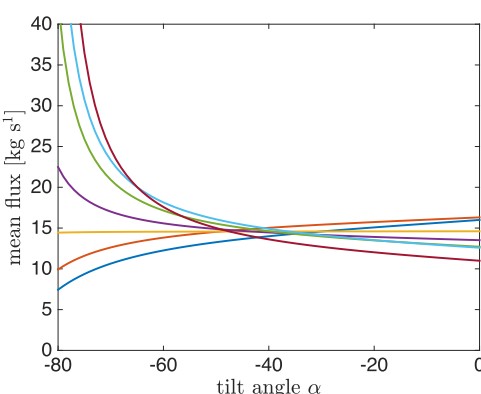

**Figure 10.** Mean fluxes of 7 different cross sections of the measurement data set in dependence on the angle correction. Each cross section is plotted in a different colour. With the a priori knowledge that the mean fluxes should be the same on time scales of hours, the plume inclination can be estimated to 40 degree with an uncertainty of 5 degree. Knowing the orientation of the camera setup this information can be used to determine the plume propagation direction.

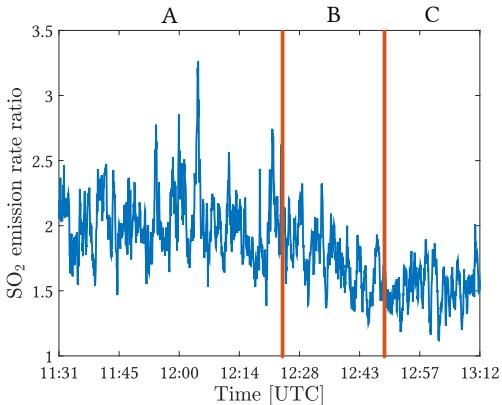

**Figure 11.** Observation of a wind direction change using the apparent flux ratios of two different cross sections of the plume which were corrected for the perspective. Depending on the propagation velocity, it is possible to determine direction changes on the time scale of minutes. For every inclination towards the image plane, the ratio of the fluxes is unique. On this data set, there occurred a direction change after phase A at 12:26 UTC. The wind direction in phase A was $281 \pm 5$ degree. The direction change of about 20 degree in phase B took approximately 23 minutes. A new stable propagation direction of $301 \pm 5$ degree established in phase C at 12:49 UTC.