# Peer review of "Plume Propagation Direction Determination with SO2 Cameras"

_Atmospheric Measurement Techniques, 2016_

## Referee Comment (RC1) · Anonymous Referee #1 · 26 Nov 2016

The paper by Klein et al. attempts to clarify geometric considerations when observing volcanic plumes by an SO$_2$ camera system. If the plume has a velocity component toward or away from the camera, derivation of instantaneous SO$_2$ fluxes can suffer from severe errors if this velocity component is neglected. The paper estimates the related errors based on geometric considerations and it recommends a method how to correct for it in real measurements. The latter is based on the assumption of a constant mean SO$_2$ flux over an (undefined) period of time.

The question addressed by the paper is technical and basic. It might be suitable for publication in AMT since it could provide citeable documentation of a methodological problem that could be used by others working in the field. However, the authors should invest in reworking the entire manuscript for clarity. Since the

main benefit of the study is clarity of documentation, it is essential to use precise and unambiguous language.

Comments:

1. I do not understand the method described in section 3:

- How important is the zenith-looking DOAS instrument? Are there any input variables (e.g. plume center distance) derived from it? If so, this should be made clearer (including the abstract and conclusion) since it implies the necessity to operate a zenith looking DOAS instrument in addition to the camera.

- What is the assumption of "constant $SO_2$ flux" (L220) and "mean flux" (L192) exactly? Do you mean that "$\Phi$" calculated through equation (1) and then, averaged over a certain period must be the same for all camera pixels? What is the averaging period?

- Is the wording "$SO_2$ flux" (equation (1), units molec/s) and "$SO_2$ emission rate" (e.g. Fig. 9, units kg/s) interchangeable for your purposes? The use of different terms for the same quantity might be confusing.

- What is "CD" (acronym never defined)? Is it the column density along the line of sight? Why not use your symbol S or s?

- L238: "This result is nicely comparable with the result from the traverse measurements." Are these traverse measurements shown anywhere? Replace "nicely" by a quantitative measure.

- Doesn't Fig. 11 show that your correction only works if wind direction does not

change? Generally, I have a hard time to follow the reasoning in section 3.2. Please extend and clarify or remove.

- Caption Fig. 11: "corrected for the perspective". What do you mean?

2. Given that the main benefit of the paper is technical documentation, it should clearly mark the range of applicability. There should be a discussion on "finer" effects and their consequences, e.g. radiative transfer issues, inhomogeneous plumes, bent plumes, several plumes partially masking each other, large $SO_2$ optical thicknesses.

3. The paper several times quotes the "compensation effect" (e.g. L252). A key motivation of the work is to show that the "compensation effect" does not work. It might be useful for the general (non-expert) reader to add some explanation and discussion on this effect. Doesn't Fig. 4 show that the compensation effect works perfectly for the telecentric "approach"?

Other comments:

Abstract, L16: "Here we propose a new method . . ." Isn't the new method based on the assumption of constant mean $SO_2$ fluxes? I would argue that this assumption is utterly important to mention, in particular since the goal is to study $SO_2$ flux variability (L25).

L44: SO2 -> $SO_2$

L47: "Future developments promise further improvements". Remove, devoid of any content.

L50: optical density -> absorption optical density

L52,55: light extinction due to scattering by aerosols and molecules

L57, 58: While not central to the paper, it might be worthwhile mentioning typical assumptions: small ODs, absorption cross section that are independent of ambient pressure and temperature?

L95: Put a reference to Fig. 1 already here.

L173: "(consisting of only one large pixel)" I do not understand this comment. Please explain.

L174: Explain what orthographic, telecentric and perspective projections are.

L232: CD: acronym not defined.

Fig. 3: "plume diameter" -> plume distance (and diameter)

Fig. 8: Why is the FOV angle called "alpha"? The text calls it "gamma", while "alpha" is the inclination angle.

Fig. 9: Caption: $SO_2$ fluxes, Figure y-axis: $SO_2$ emission rates. Are they the same?

Fig. 10: "Mean flux" -> Is it the same quantity as the $SO_2$ emission rates in Fig. 9? What is the averaging period?

---

## Referee Comment (RC2) · Anonymous Referee #2 · 19 Dec 2016

The paper by Klein et al. deals with geometric effects which need to be considered for a correct flux determination of the volcanic plume. The assumption of flux continuity can be used to determine the plume propagation direction and to infer the flux corrected for effects of perspective / projection. I agree to referee #1 that the presentation is somewhat difficult to follow and the suggested corrections would improve clarity. Please explain equation 1 with greater care: what is the meaning of the summation index i? I assume that the summation is performed just along a column of pixels, not over all pixels of the array. Does the equation assume that pixels are quadratic in shape? What is the "diameter" of a quadratic pixel?

I do not understand the discussion of the telecentric projection. The assumption of a telecentric projection might be a useful intermediate step for demonstrating the additional effects of a perspective projection, but the statement " . . . a perspective correction

is more common in SO2 camera . . . setups." remains obscure. For a telecentric projection in object space, you need an optical aperture of the size of the object – I can hardly imagine that any SO2 camera has ever been built that realizes a telecentric projection in object space of a volcanic plume.

You mention the optical flow algorithms by Kern et al.. Advanced methods of flow reconstruction have been developed which use the condition of continuity more explicitely. I think it would be good to cite this approach and to investigate whether such a rigorous physical model (taking into account the projection effects you point out) would allow for the optimum determination of plume parameters. You might refer to the work of Stremme et al., AMT, 2012.

———————————————————

---

## Referee Comment (RC3) · Anonymous Referee #3 · 20 Dec 2016

Plume Propagation Direction Determination with SO2 Cameras

Klein et al

Review

The paper describes an algorithm which can be used to convert time-evolving 2D imagery from an SO2-sensitive camera into SO2 emission rates. The authors apply a full geometric treatment to the sounding problem, and evaluate the errors associated with some commonly used simplifying approximations, including the assumption that emission rate estimates are insensitive to plume direction for a small angular FOV.

Further they note that if emission rate is assumed constant, the actual direction of the plume relative to the viewing direction can also be determined.

[Figure]

The papers concludes with a demonstration of an application to real data acquired from Mt Etna.

General Comments

I am not familiar with this area of remote sensing but I am a little surprised that independent information on the wind/plume direction is so scarce: no (micro) weather station telemetry, meteorological forecasts, or even a second observing camera viewing the same plume from a different location? All seem simpler alternatives than trying to derive wind direction from the 2D observations from a single site.

A second, perhaps poorly informed, comment is that I assume that there must be some irregularity in the SO2 emission otherwise it would not be possible to determine (angular) movement of the plume across the camera view. Is this consistent with the assumption of constant emission rate required to determine the plume direction.

Minor Typographical/Gramamtical comments:

- 'degree' is commonly used where 'degrees' is correct.

- 'extension' is commonly used where 'extent' is correct.

- L.146 'alpha' rather than 'apha' (although I would suggest using the symbol instead in alpha<0 and alpha>0).

- Fig 3 caption: 'plume extent in x-direction'.

- Captions: 'in dependence of' is perhaps better expressed as 'as a function of', or rephrased along the lines of 'showing the dependence upon' or 'showing the variations with'.

———

---

## Author Comment (AC1) · 28 Jan 2017

The paper by Klein et al. attempts to clarify geometric considerations when observing volcanic plumes by an SO2 camera system. If the plume has a velocity component toward or away from the camera, derivation of instantaneous SO2 fluxes can suffer from severe errors if this velocity component is neglected. The paper estimates the related errors based on geometric considerations and it recommends a method how to correct for it in real measurements. The latter is based on the assumption of a constant mean SO2 flux over an (undefined) period of time.

The question addressed by the paper is technical and basic. It might be suitable for publication in AMT since it could provide citeable documentation of a methodological problem that could be used by others working in the field. However, the authors should invest in reworking the entire manuscript for clarity. Since the main benefit of the study is clarity of documentation, it is essential to use precise and unambiguous language.

Answer to Referee Comment 1: We do not agree with the formulation "documentation of a methodological problem" since our approach is new and has not been used before. In addition, our new approach allows to derive an additional parameter from the SO2 camera data sets, the wind direction at the position of the volcanic plume. However, we do agree that we need to improve the clarity of our manuscript. We made many changes to the manuscript to improve the clarity of presentation as described in detail below.

Comments:

Referee comment 2: I do not understand the method described in section 3:
How important is the zenith-looking DOAS instrument? Are there any input variables (e.g. plume center distance) derived from it? If so, this should be made clearer (including the abstract and conclusion) since it implies the necessity to operate a zenith looking DOAS instrument in addition to the camera.

Answer to Referee Comment 2: The zenith-looking DOAS instrument was only used during this particular measurement campaign in order to provide an independent verification of the wind direction derived from our new perspective correction algorithm. In future SO2 camera measurements alone will deliver sufficient information and additional measurements by a zenith-looking DOAS instrument will not be necessary. To clarify this statement, as proposed by the referee, we added a small change to the manuscript (printed in bold face):
**»During the measurement campaign, in addition to the SO$_2$ camera measurements, measurements were taken by a car based DOAS instrument pointing to the zenith and traversing the plume (see e.g. McGonigle et al., 2002, Galle et al., 2003). The center of the plume can be found by evaluating the SO$_2$ column density and determining the location with the maximum values. Thus, the wind direction could be estimated, giving an inclination of the plume of about 38 degrees. This information was then used to verify the applicability of the developed algorithm to the SO$_2$ camera data sets. «** (**L236ff)**

Referee Comment 3: What is the assumption of "constant SO2 flux" (L220) and "mean flux" (L192) exactly? Do you mean that "Φ" calculated through equation (1) and then, averaged over a certain period must be the same for all camera pixels? What is the averaging period?

Answer to Referee Comment 3: We fully agree with the referee that we need to improve the wording. What we indeed intended to say, is that the SO2 amount in the air parcel that travels through the field of view does not change due to chemical reactions on the time scales of interest (i.e. the time required for the plume to cross the field of view of the camera) and that therefore SO2 acts as a conservative tracer. The second necessary assumption is that the advective transport of SO2 exceeds the turbulent transport of SO2 in the plume. This usually holds true, e.g. Kern, 2009 finds that for typical conditions the turbulent flux is only 0.2% of the advective flux. Here are our corrections to the manuscript:

»**If we want to determine the plume propagation direction, we can measure the SO₂ flux for a given distance at different positions of the plume. If the apparent SO₂ flux is the same at different positions in the plume, the plume lies within the image plane. Otherwise we observe an apparent gradient in the measured flux across the image, which, however, contains the plume propagation direction information of the plume. Dividing the measured fluxes by the respective deviations for the investigated pixel columns for every possible tilt angle alpha and minimizing the observed gradient yields the information about the mean plume propagation direction during the time period the parcel needed to move across the field of view. The assumption that the SO₂ flux is conserved can be made since the mean lifetime of SO₂ in the troposphere usually is of the order of several days (e.g. Eisinger and Burrows, 1998) while the typical time for the plume to cross the field of view of the camera is of the order of a few minutes. Note that the SO₂ flux originating from the volcano should vary for our method to work. The second necessary prerequisite is that the advective transport is larger than the turbulent transport of the plume parcel. Kern, 2009 showed that usually the turbulent transport only contributes to about 0.2 percent of the gas transport in the plume**.« (L193ff).

Referee Comment 4: Is the wording "SO2 flux" (equation (1), units molec/s) and "SO2 emission rate" (e.g. Fig. 9, units kg/s) interchangeable for your purposes? The use of different terms for the same quantity might be confusing.

Answer to Referee Comment 4: We agree with the reviewer that we should use the same term throughout instead of two. Indeed the two terms are interchangeable since we know the mass of SO2 molecules.
We changed all occurrences of "SO2 emission rate" to "SO2 flux"

Referee Comment 5: What is "CD" (acronym never defined)? Is it the column density along the line of sight? Why not use your symbol S or s?

Answer to Referee Comment 5: We agree with the reviewer that it is not necessary to write CD, we corrected it in the manuscript to "column density"

- L238: "This result is nicely comparable with the result from the traverse measurements." Are these traverse measurements shown anywhere? Replace "nicely" by a quantitative measure.
Answer to Referee Comment 6: The traverse measurements are not shown in this paper. However, only the wind direction determined using the traverse measurements are used to compare the result of our algorithm to the measured wind direction. As suggested by the referee we replaced the term "nicely" by a quantitative measure as follows:
»**This result agrees with the result from the traverse measurements within an error of ±2 degrees.**«(L245f)

Referee Comment 7: Doesn't Fig. 11 show that your correction only works if wind direction does not change? Generally, I have a hard time to follow the reasoning in section 3.2. Please extend and clarify or remove.

Answer to Referee Comment 7: Figure 11 shows that the correction even works on time scales of minutes. We agree that it was hard following the reasoning since our initial wording of the "constant mean flux" needed to be corrected. We further improved by adding the following text to the manuscript:

»Figure 11 shows a change in the ratio between the apparent $SO_2$ flux determined in two different positions of the plume within the
FOV of the camera. During 2 hours of measurement between 11:32 - 13:23, on 9th July 2014, the wind direction was stable for about one hour (A) for this period the ratio between the two flux measurement positions was mostly constant, except for slight fluctuations. Then the inclination angle changed by about 20 degrees, which we attribute to a change of the wind direction (B). In this phase of the observation the ratio between the two flux measurements is decreasing. Later, the new wind direction stabilizes in (C), with a new constant ratio between the two flux measurements. The $SO_2$ plume also exhibits a slight contribution due to turbulent transport. The turbulent transport usually contributes less than 0.2 percent to the propagation of the plume (Kern, 2009).«(L250ff)

Referee Comment 8: Caption Fig. 11: "corrected for the perspective". What do you mean?

Answer to Referee Comment 8: The reviewer is right, unfortunately, a small error occurred and the wording "corrected for the perspective" is falsely used in this context. We removed it in the manuscript.

Referee Comment 9: Given that the main benefit of the paper is technical documentation, it should clearly mark the range of applicability. There should be a discussion on "finer" effects and their consequences, e.g. radiative transfer issues, inhomogeneous plumes, bent plumes, several plumes partially masking each other, large SO2 optical thicknesses.

Answer to Referee Comment 9: As we pointed out in our answer to referee comment 1, this technical documentation is about a new method and so far the total range of its applicability is not yet fully investigated. Radiative transfer issues (which also include the case of large SO2 optical thicknesses) have been neglected so far. These issues have so far not even sufficiently been solved for even scanning DOAS systems. Bent plumes (i.e. plumes where the wind direction changes within the field of view) can be detected with this method as long as the bent exceeds more than a few pixel columns in extend. Inhomogeneous plumes contribute to rather good measurement conditions since it improves the velocity determination of the plumes. The case of plumes partially masking each other is indeed a situation where the algorithm may not work. We added these proposed comments to the manuscript for clarity:

»While the proposed algorithm is applicable to straight and bent plumes (i.e. plumes where the wind direction changes within the field of view), so far the combined effects of this study and radiative transfer issues have not yet been addressed. Additionally, if several plumes are masking each other, the proposed algorithm may not work.«(L276ff)

Referee Comment 10: The paper several times quotes the "compensation effect" (e.g. L252). A key motivation of the work is to show that the "compensation effect" does not work. It might be useful for the general (non-expert) reader to add some explanation and discussion on this effect. Doesn't Fig. 4 show that the compensation effect works perfectly for the telecentric "approach"?

Answer to Referee Comment 10: Part of the motivation of this work is indeed that the compensation effect does not work on larger field of views. However, we also show (in section 2) that for small FOVs and for small inclination angles of the plume it can well be assumed to hold. The telecentric approach is mentioned in the manuscript for the sake of completeness since it is only a theoretical consideration. No telecentric $SO_2$ camera has been built so far.

Other comments:

Referee Comment 11: Abstract, L16: "Here we propose a new method ..." Isn't the new method based on the assumption of constant mean SO2 fluxes? I would argue that this assumption is utterly important to mention, in particular since the goal is to study SO2 flux variability (L25).

Answer to Referee Comment 11: We agree with the referee that, as also mentioned in our answer to referee comment 3, we need to improve the description of the prerequisites of the method. As described in Answer on the Referee Comment 1 we made several changes to the description in the manuscript to make clear that not a constant SO2 flux is required but SO2 must be a conservative tracer on the time scale of a few minutes (i.e. for the time the plume needs to traverse the field of view of the camera).

Referee Comment 12: L44: SO2 -> SO2
Answer to Referee Comment 12: We corrected the error.

Referee Comment 13: L47: "Future developments promise further improvements". Remove, devoid of any content.
Answer to Referee Comment 13: We removed the sentence as suggested.

Referee Comment 14: L50: optical density -> absorption optical density

Answer to Referee Comment 14: We are referring to the extinction optical density and added the word "extinction" to the manuscript for clarity:
**»The SO$_2$ camera is a UV sensitive camera utilizing one or more band-pass interference filters to measure the extinction optical density (OD) of SO$_2$. «(L47ff)**

Referee Comment 15: L52,55: light extinction due to scattering by aerosols and molecules

Answer to Referee Comment 15: The SO2 camera measures the light extinction due to absorption by SO2 and due to scattering by aerosols.

Referee Comment 16: L57, 58: While not central to the paper, it might be worthwhile mentioning typical assumptions: small ODs, absorption cross section that are independent of ambient pressure and temperature?

Answer to Referee Comment 16: Small ODs are not necessary for measurements with SO2 cameras and for the use of this algorithm. While we agree that the (differential) absorption cross section of some gases can be sensitive to ambient pressure and temperature, it does not contribute to the major challenges of SO2 camera measurements.

Referee Comment 17: L95: Put a reference to Fig. 1 already here.

Answer to Referee Comment 17: We agree with the referee and added a reference to the figure in the manuscript.

Referee Comment 18: L173: "(consisting of only one large pixel)" I do not understand this comment. Please explain.
Answer to Referee Comment 18: We agree with the referee that this wording may be unclear. We wanted to point out, that each half of the image plane would be handled as single detector plate to provide a better understanding towards the transition to a many pixel camera detector array. We

removed the remark in the brackets and adjusted the text as shown in Answer to Referee Comment 19.

Referee Comment 19: L174: Explain what orthographic, telecentric and perspective projections are.

Answer to Referee Comment 19: We added explanations for orthographic and perspective projection to the manuscript as follows:
**»The graphs show the deviations for half of the image plane from its center to an angle gamma/2 of 12 degree where half of the image plane is assumed to be a single detector pixel. An orthographic projection (produced by a telecentric optical setup) leads to the blue lines in Fig. 3. Its characteristic is that an object is projected the same size independent on its distance to the camera. The red lines in Fig. 3 represent the deviations due to a perspective projection that is used in $SO_2$ camera measurement setups. The projected size of an object is dependent on the distance for a perspective projection.«(L172ff)**

Referee Comment 20: L232: CD: acronym not defined.
Answer to Referee Comment 20: We corrected to "column densities" and removed "CD", since the acrony only occurs twice.

Referee Comment 21: Fig. 3: "plume diameter" -> plume distance (and diameter)
Answer to Referee Comment 21: We corrected to "plume distance".

Referee Comment 22: Fig. 8: Why is the FOV angle called "alpha"? The text calls it "gamma", while "alpha" is the inclination angle.
Answer to Referee Comment 22: We agree with the referee. This is a typo. We corrected to "gamma".

Referee Comment 23: Fig. 9: Caption: SO2 fluxes, Figure y-axis: SO2 emission rates. Are they the same?
Answer to Referee Comment 23: Since the mass of an SO2 molecule is known, we could use the quantities interchangeable, see answer to author comment 4. For a better understanding we only will use flux in the manuscript.

Referee Comment 24: Fig. 10: "Mean flux" -> Is it the same quantity as the SO2 emission rates in Fig. 9? What is the averaging period?
Answer to Referee Comment 24: We adjusted the wording to be consistent.
Referee Comment 25: The paper by Klein et al. deals with geometric effects which need to be considered for a correct flux determination of the volcanic plume. The assumption of flux continuity can be used to determine the plume propagation direction and to infer the flux corrected for effects of perspective / projection. I agree to referee #1 that the presentation is somewhat difficult to follow and the suggested corrections would improve clarity. Please explain equation 1 with greater care:

what is the meaning of the summation index i? I assume that the summation is performed just along a column of pixels, not over all pixels of the array. Does the equation assume that pixels are quadratic in shape? What is the "diameter" of a quadratic pixel?

Answer to Referee Comment 25: We agree with the referee, that we need to explain equation 1 with greater care. We added further explanations to the manuscript. However, it is not necessary that the pixels are quadratic in shape. Knowing the distance of the plume and knowing the shape of the pixels every extent in every direction a pixel represents can be calculated.

**»Here, v is the propagation velocity of the plume perpendicular to the viewing direction, $h_i$ is a side length of a pixel at the distance of the plume and $S_i$ denotes the $SO_2$ column densities of each respective pixel. The summation of the length of every pixel in the plume transect gives the overall diameter of the plume. The summation over every column density of these pixels gives the column density of the complete transect. «(L85ff)**

Referee Comment 26: I do not understand the discussion of the telecentric projection. The assumption of a telecentric projection might be a useful intermediate step for demonstrating the additional effects of a perspective projection, but the statement " . . . a perspective correction is more common in SO2 camera . . . setups." remains obscure. For a telecentric projection in object space, you need an optical aperture of the size of the object – I can hardly imagine that any SO2 camera has ever been built that realizes a telecentric projection in object space of a volcanic plume.

Answer to Referee Comment 26: We agree with the referee. We intended to use the telecentric approach exactly for the mentioned purpose, as a demonstration of the additional effects of a perspective projection. We changed the manuscript for a clearer description of the intended purpose:

**»The red lines in Fig. 3 represent the deviations due to a perspective projection that is used in $SO_2$ camera measurement setups.«(L175ff)**

Referee Comment 27: You mention the optical flow algorithms by Kern et al.. Advanced methods of flow reconstruction have been developed which use the condition of continuity more explicitely. I think it would be good to cite this approach and to investigate whether such a rigorous physical model (taking into account the projection effects you point out) would allow for the optimum determination of plume parameters. You might refer to the work of Stremme et al., AMT, 2012.

Answer to Referee Comment 27: We agree with the referee. It is indeed very beneficial if one combines our corrections with advanced plume velocity determination algorithms. In fact we already used modern optical flow algorithms (e.g. the one described by Peters et al., 2015, that make use of the continuity condition) on some heterogeneous plume data sets at Stromboli volcano (Klein, Master's Thesis 2015). However, including this approach in the present manuscript would need a detailed explanation of the flow reconstruction methods und we decided that it would exceed the framework of this manuscript. But we agree that it would improve the manuscript if we added this outlook to our conclusion:

**»Recent improvements in the velocity determination of volcanic plumes using image processing methods like optical flow algorithms can be combined with the proposed perspective correction method for a robust $SO_2$ flux determination.« (L276ff)**
The paper describes an algorithm which can be used to convert time-evolving 2D imagery from an SO2-sensitive camera into SO2 emission rates. The authors apply a full geometric treatment to the sounding problem, and evaluate the errors associated with some commonly used simplifying approximations, including the assumption that emission rate estimates are insensitive to plume direction for a small angular FOV.

Referee Comment 28: Further they note that if emission rate is assumed constant, the actual direction of the plume relative to the viewing direction can also be determined.
Answer to Referee Comment 28: We repeat that in addition, our new approach allows to derive an additional parameter from the SO2 camera data sets, the wind direction at the position of the volcanic plume. As pointed out by the referees, we did not unambiguously describe the prerequisite for the plume direction determination. We changed the manuscript for more clarity regarding this point as described in author comment 3.

The papers concludes with a demonstration of an application to real data acquired from Mt Etna.

General Comments

Referee Comment 29: I am not familiar with this area of remote sensing but I am a little surprised that independent information on the wind/plume direction is so scarce: no (micro) weather station telemetry, meteorological forecasts, or even a second observing camera view- ing the same plume from a different location? All seem simpler alternatives than trying to derive wind direction from the 2D observations from a single site.

Answer to Referee Comment 29: It is indeed surprisingly difficult to determine the wind velocity and direction at the specific position of the plume. Wind information close to the ground as given by a weather station do usually not represent the wind conditions in the respective plume height due to orographic influence of the surrounding mountains (and volcanoes are usually mountains) and due to the Ekman twist. Therefore, a method to determine the true wind data at the position of the plume should be very welcome.

Referee Comment 30: A second, perhaps poorly informed, comment is that I assume that there must be some irregularity in the SO2 emission otherwise it would not be possible to determine (angular) movement of the plume across the camera view. Is this consistent with the assumption of constant emission rate required to determine the plume direction.

Answer to Referee Comment 30: We agree that for a good velocity estimation irregularity in the SO2 emission is a necessary prerequisite. We also agree that our wording of the constant emission rate is misleading, as already mentioned in referee comment 3.

Minor Typographical/Gramamtical comments:

Referee Comment 31: 'degree' is commonly used where 'degrees' is correct.

Answer to Referee Comment 31: We agree with the referee and corrected the errors in our manuscript.

Referee Comment 32: 'extension' is commonly used where 'extent' is correct.
Answer to Referee Comment 32: We agree with the referee and corrected the errors in our manuscript.

Referee Comment 33: L.146 'alpha' rather than 'apha' (although I would suggest using the symbol instead in alpha<0 and alpha>0).
Answer to Referee Comment 33: We agree with the referee and corrected the errors in our manuscript.

Referee Comment 34: Fig 3 caption: 'plume extent in x-direction'.
Answer to Referee Comment 34: We agree with the referee and corrected the errors in our manuscript.

Referee Comment 35: Captions: 'in dependence of' is perhaps better expressed as 'as a function of', or rephrased along the lines of 'showing the dependence upon' or 'showing the variations with'.
Answer to Referee Comment 35: We agree with the referee and corrected the errors in our manuscript.